# Addressing a Rule of Thumb: Modeling the Effects of Meteorological Conditions on Prescription of Antimicrobials in Aquaculture

Kasper Rømer Villumsen,[a] Anders Miki Bojesen[a]

[a]Department of Veterinary and Animal Sciences, Faculty of Health and Medical Sciences, University of Copenhagen, Copenhagen, Denmark

**ABSTRACT** Meteorological phenomena such as warm summer temperatures and increased numbers of sunlight hours have repeatedly been hypothesized to be associated with infectious diseases and an increased prescription of antimicrobial compounds in Danish aquaculture. To address this hypothesis, we prepared statistical models incorporating aquaculture production data from Danish Aquaculture, detailed records on prescription of antimicrobials from the Danish VETSTAT program, and meteorological data from 2001 to 2019 from the Danish Meteorological Institute. Separate series of models were made and refined for land-based and marine production, respectively. For both production forms, the models identify summer sunlight hours as having a significant influence on antimicrobial use. In addition to summer sunlight, spring sunlight and water temperature were integral, although not statistically significant, parameters when modeling antimicrobial use in marine production. Although the extensive availability and accuracy of relevant data are associated with Danish production, we believe the results allow for more general conclusions on the influence of meteorological parameters on outbreaks of bacterial pathogens in international aquaculture. Such insights could have a substantial impact on prophylactic strategies, fish husbandry, and our understanding of how increasing temperatures may affect future antimicrobial usage in the global aquaculture industry.

**IMPORTANCE** Global aquaculture production has been rapidly increasing for decades and is set to play a pivotal role in feeding a growing human population. Along with the growth in aquaculture production, the annual global use of antimicrobials is estimated to increase by one-third between 2017 and 2030. Using detailed antimicrobial prescription records as a proxy for outbreaks, we were able to evaluate the effects of a variety of meteorological parameters through statistical modeling. Our results lend scientific support to informal observations from the industry, but more importantly, this study provides novel, essential information on the importance of abiotic factors that can, in turn, lead to improved prophylactic efforts and thus help to reduce antimicrobial use in global aquaculture.

**KEYWORDS** aquaculture, climate, statistical modeling, antimicrobial agents, veterinary microbiology

Historically, the recorded annual use of antimicrobial (AM) compounds in Danish aquaculture has undergone noticeable fluctuations. The fluctuations have consistently been attributed to air temperatures where increased use of AM treatments was attributed to high summer temperatures, yet only limited documentation has been reported in support of this assumption (1, 2). Over time, this apparent connection between weather and AM treatments has led to a common assumption among laypeople and professionals that warm summers lead to increased use of AM due to higher infectious pressure. To the best of our knowledge, a potential correlation between

Address correspondence to Anders Miki Bojesen, miki@sund.ku.dk.

The authors declare no conflict of interest.

meteorological phenomena and AM use has not been directly addressed. Thus, inspired by the observed variation in antimicrobial use in Danish aquaculture, the current investigation aimed at addressing whether various meteorological factors could be generally associated with the prescription of AM in aquaculture.

A detailed investigation of correlations between AM treatments and meteorological data requires access to detailed and accurate data. In Denmark, treatments with AM, whether for human, clinical, or veterinary use, require prescriptions, which are recorded by a central government agency. Since 1995, AM use has been made available through the Danish integrated Antimicrobial Resistance Monitoring and Research Program (DANMAP) (2). Annual registration reports allow separate tracking of AM prescriptions in veterinary practices (VETSTAT), including separate records for aquaculture, starting in 2000. The detailed recording practices offered by VETSTAT and the resulting data allow a unique insight into the dynamics of both total AM usage, as well as compound-specific use within the aquaculture sector (3). In combination with meteorological data, these uniquely detailed AM prescription records enable investigations of factors associated with annual fluctuations in AM usage. While the analyses presented in the current study are based on Danish meteorological and aquaculture production data, the insights gained and the potential underlying mechanistic explanations for the observed variation in AM use are expected to be generally applicable to aquaculture at similar latitudes.

To set the framework for the data foundation, a brief description of Danish aquaculture production is given here. Total Danish aquaculture production consists of a land-based part and a marine part. As of 2019, land-based production included 118 traditional farms, employing a flowthrough water supply, as well as 33 so-called model farms based on extensive recirculation of water, while marine production consisted of 19 marine farms based on net pens. In addition to this, 37 were designated "other farm types," covering eel farms, land-based seawater farms, and other farm setups that do not fall into one of the major categories (4). Land-based production accounts for approximately three-quarters of the total production (Fig. 1). The primary species produced is rainbow trout (*Oncorhynchus mykiss*), accounting for approximately 85% of the total production (5, 6). The land-based production typically relies on local watercourses or groundwater, sometimes with high levels of recirculation, whereas marine farming is subject to more environmental variation. Danish waters are generally shallow, with marine production typically taking place in waters of 5 to 15 m depth and with average surface currents of 10 to 20 cm/second (7, 8). Production near the coast at similar depths falls under the category of coastal farming (9), and the described current speeds are designated "little exposure" (0 to 30 cm/second) for sea cage farming by Standards Norway, making Danish production areas relatively shallow with low water current speeds (10). Given these limited depths and current speeds, a notable effect of various weather conditions on the water column, and thus the immediate, marine production environment, must be considered plausible.

Husbandry practices include extensive vaccination programs, with estimated vaccination coverages of approximately 90% for both immersion vaccination (enteric redmouth disease, *Yersinia ruckeri*), as well as injection vaccination (furunculosis and vibriosis, *Aeromonas salmonicida* and *Vibrio anguillarum*, respectively) (11). Despite prophylactic measures, however, treatments with AM are still necessary.

In the present study, we considered two rules of thumb relating to AM use, hypothesizing that (i) there is a correlation between warmer summers and higher AM usage, and (ii) rapid increases in temperature and increased solar exposure promote outbreaks of bacterial infection and, as a result, increased AM usage. We tested the hypotheses by investigating whether variations in prescription of antimicrobials in Danish aquaculture were associated with variations in meteorological parameters.

Specifically, we investigated the influence of the following parameters: air temperature, sunlight, marine water temperature, and monthly increases in both air and marine water temperature during spring and summer months, respectively, on the annual prescription of antimicrobials.

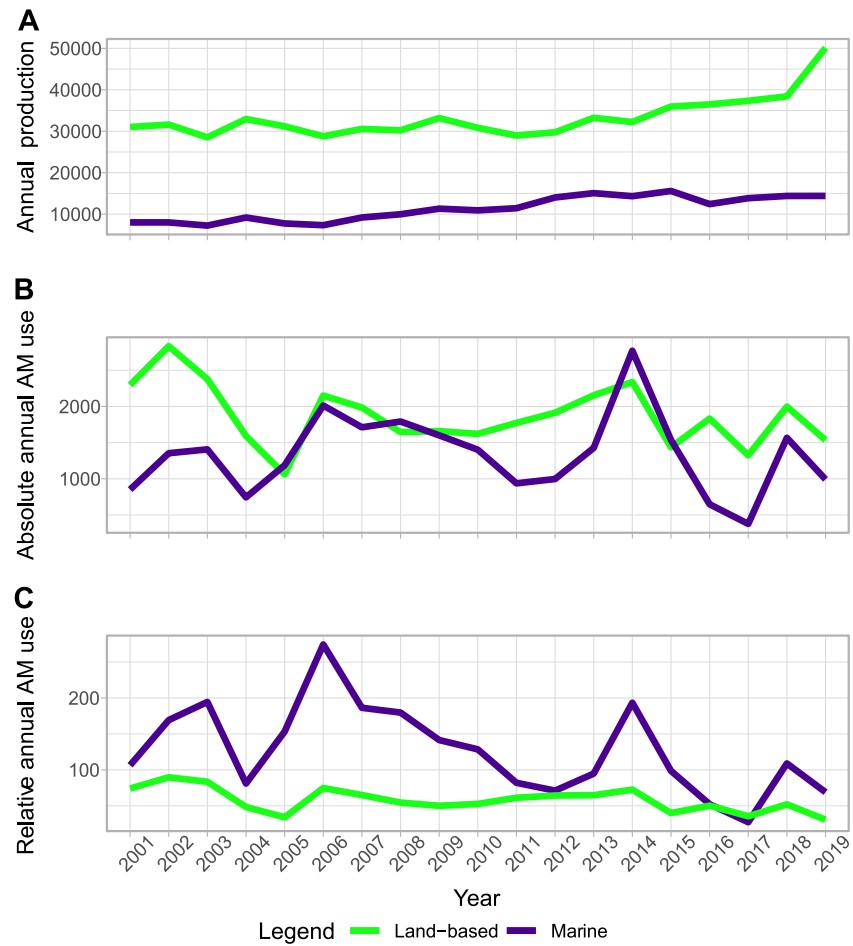

**FIG 1** Overview of production, AM use, and AM use relative to production for Danish aquaculture from 2001 to 2019. (A to C) Annual production (metric tons) (A), absolute AM use (kilograms) (B), and relative AM use (grams/metric ton) (C) in land-based (green) and marine (blue) production.

Considering their different characteristics, relative production totals, as well as the fact that not all data parameters applied to both land-based and marine production forms, the land- and marine-based systems were addressed separately. With the aim of providing mechanistic insights into the roles of meteorological parameters on variation in AM usage in aquaculture, families of iterative generalized linear models were set up to find the most influential drivers of this variation and combine them into parsimonious, final models for each production form.

## RESULTS

**Variation in aquaculture production and AM usage data.** Annual production, as well as absolute and relative use of AMs for both land-based and marine aquaculture in Denmark between 2001 and 2019, is shown in Fig. 1 (Data is available upon request).

Throughout the study period, relatively consistent production totals were observed, with mean totals of 33,230 ($\pm$5,023 standard deviation [SD]) and 11,283 ($\pm$2,920 SD) metric tons for land-based and marine aquaculture, respectively. The standard deviations imply a variation of 15.1% and 25.9% of the annual mean production values for land-based and marine production, respectively. With sulfonamides-trimethoprim as the predominant treatment, followed by quinolones and amphenicols (Fig. S1), the total annual use of AM in aquaculture from 2001 to 2018, as reported by DANMAP, fluctuated between 1,697 kg and 5,116 kg of active compound (1, 2, 12–28). When

addressing the absolute use of AM for each production form independently, considerable variation was observed throughout the study period, with the marine production at times exceeding the AM use of the numerically greater land-based production (Fig. 1B). For the land-based production, a mean annual AM usage of 1,870 ($\pm$426.7 SD) kg was observed, with a variation equal to 22.8% of the mean annual use. For marine production, the mean annual AM usage was 1,332 kg ($\pm$550 SD), with a variation equal to 41.3% of the mean. When normalizing the AM usage by calculating the AM usage (grams active substance) per weight (metric tons) produced, this pattern of variation remained, and further, the relative AM usage was consistently higher for marine production, except for 2016 and 2017 (Fig. 1C).

The data on AM usage are based on recorded prescriptions and thus directly reflect the need for treatments of outbreaks during any given year. The available vaccine sales figures, however, merely reflect sales figures within a given year, not the actual use. It was thus not possible to tell whether the vaccines sold were, in fact, used within the year of purchase. Since the vaccine data would then introduce an unknown level of uncertainty, they were not included in the analyses.

**Meteorological data.** Data obtained from the Danish Meteorological Institute (DMI) are shown in Table S1 and plotted in Fig. S2 and indicate differing degrees of variation in annual observations among all included parameters.

**Modeling variation in the use of antimicrobials. (i) Land-based production.** The water temperatures represent marine measurements and therefore are not included when modeling land-based AM usage, whereas data variables concerning sunlight, air temperature, absolute AM use, and land-based production were included.

Before any models were set up, the eligible variables were investigated for any substantial correlations among them, as such correlations could impact model behavior, as well as the conclusions made from those models. Based on this correlation analysis, it should be noted that summer air temperature and summer sunlight (Pearson's correlation coefficient [PCC] = 0.74), as well as summer sunlight and AM usage, appeared to be correlated (PCC = 0.75). A Pearson correlation network of the variables is shown in Fig. 2, providing a graphical representation of the intercorrelations of the variables.

Having addressed the potential of correlation between variables, the initial "full" model of AM use in land-based production was constructed by fitting all eligible predictor variables to the response variable (annual AM use). Based on the initial results of the "full" model starting point, iterative, manual refinement of the model was then applied, which resulted in the sequence of models given in Table 1. For full details on each iteration, all statistical material is available upon request.

After the seven nested iterations of the model were constructed, their respective fit to the data was compared as described in Materials and Methods. Using Akaike information criteria (AIC) as a guide, the model land 3 posted the lowest AIC and the highest AIC weight. The land 3 model incorporates summer sunlight and annual land-based production totals as predictor variables, with no interaction links. The model output for land 3 indicated that only the summer sunlight variable had a statistically significant impact on the response variable, the annual AM use ($P = 0.0011$; for definitions of null and alternative hypotheses, see Materials and Methods). When examining the model output for the remaining models, a statistically significant impact of summer air temperature variables on the response variable was found ($P = 0.003$) for the land 5 model, as well as a significant effect of production volume ($P = 0.046$). When the models are compared among each other using the evidence ratio (ER) as a measure of relative strength, land 3 is 3 times more likely to be the best model fit than land 5. When further compared to land 4, which considers summer air temperature as well as summer sunlight and results in the second-lowest AIC score, land 3 is 1.2 times more likely to be the best model fit by way of calculated ER. By comparison, the more complex models featuring >3 predictor variables and/or interaction links fit to the AM use data all resulted in higher AIC values and lower AIC weights, likely reflecting that these are overfit to the data.

As the land 3 model featured just two predictor variables in addition to the response variable, the relationships between these were plotted in three dimensions;

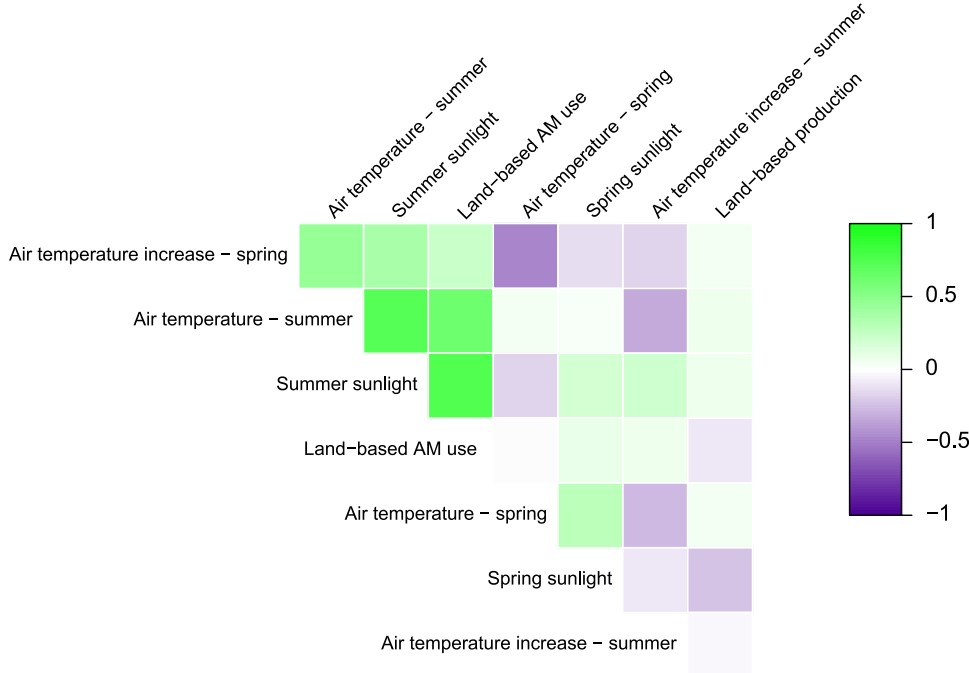

**FIG 2** Correlation matrix of data parameters for land-based production. Each comparison is represented by a colored square. Parameter correlation is by color, progressing from negative correlation (purple) through no correlation (white) to positive correlation (green).

see Fig. 3. From this graphical presentation of model parameter data, the linear relationship between summer sunlight hours and annual AM use can be observed.

To further assess the variation in the multidimensional data set, excluding the response variable, absolute AM use was further assessed by principal-component analysis (PCA; see Fig. S3) (29). Out of a total of eight principal components (PCs), the combination of the two most impactful (PC1 and PC2) explained 53% of the variation within the data set. The variable loadings, describing the correlation between a variable and PC1 and PC2 (Fig. S1), indicate that summer sunlight, summer air temperature, and spring air temperature increase contribute to the variation observed within PC1, whereas land-based production total, spring sunlight, spring air temperature, and summer air temperature increase contribute to the variation observed within PC2. Finally, the data points within the PCA plot were subdivided according to the annual AM use relative to the mean (over/under); the scattering of data points occurs mainly along the PC1 axis, driven by summer sunlight, summer air temperature, and spring air temperature increase.

**TABLE 1** Summary of land-based gamma GLM iterations[a]

| Model | Parameters | AIC | ΔAIC | AIC wt |
|---|---|---|---|---|
| Land, full | Spring temp * spring sunlight + summer temp * summer sunlight + temp increase air spring + temp increase air summer + land-based production | 283.7 | 7.7 | 0.007 |
| Land 1 | Spring temp * spring sunlight + summer temp * summer sunlight + land-based production | 281.1 | 5.1 | 0.028 |
| Land 2 | Summer temp * summer sunlight + land-based production, | 278.1 | 2.1 | 0.126 |
| **Land 3** | **Summer sunlight + land-based production** | **276.0** | **0.0** | **0.36** |
| Land 4 | Summer sunlight + summer temp + land-based production | 276.4 | 0.4 | 0.30 |
| Land 5 | Summer temp + land-based production | 278.1 | 2.1 | 0.12 |
| Land 6 | Summer temp * summer sunlight | 279.6 | 3.5 | 0.06 |

[a]Each iteration of the model fit to the annual AM use is listed with included model parameters. Asterisk separators denote that both connected parameters, as well as any interaction effects between them, are included in the model. The AIC for each model is given along with the ΔAIC (numerical difference between a given model AIC and the lowest AIC obtained). The AIC weight represents the probability that the model is the best among all iterations. The model resulting in the lowest AIC is highlighted in bold.

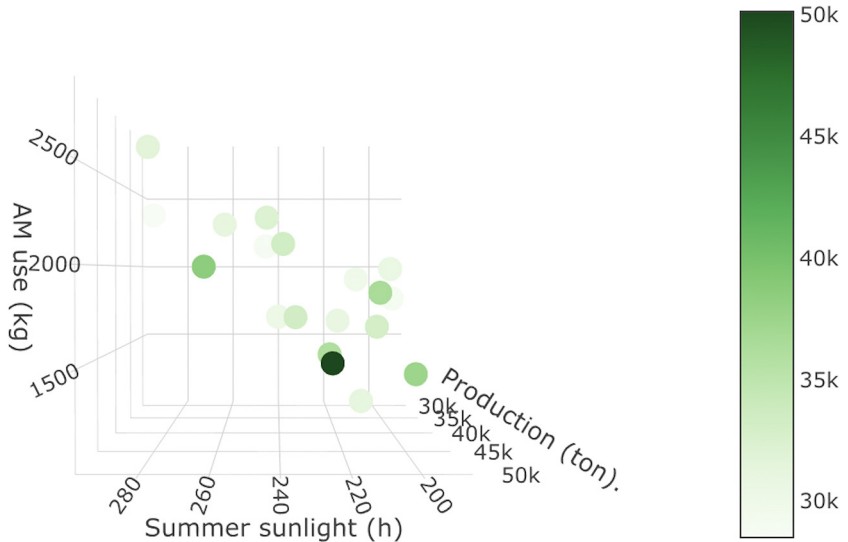

**FIG 3** Three-dimensional (3D) plot of annual values for land 3 model variables. Each dot represents a year plotted according to variable values. The additional color scale reflects the annual production use (z axis).

**(ii) Marine production.** As for the land-based production, the eligible data variables for marine production were investigated for intercorrelation before any models were constructed. For marine production, several data parameters related to air and water temperatures were found to be noticeably positively correlated (PCC ≥ 0.72), as were summer air temperature and summer sunlight (PCC = 0.74). As linear relationships, or collinearity, between predictor variables can adversely affect model performance, and as several correlations were observed between air and water temperature-related data, all data related to air temperature parameters were excluded from subsequent modeling efforts. The relationships between variables are shown in a Pearson correlation network plot of the analysis (Fig. 4).

Having addressed correlations between data variables, an initial full model was computed, fitting all remaining eligible data parameters. As for the land-based production models, the evaluation of the full model was followed by iterative manual trimming based on AIC and statistical significance in terms of model coefficients.

The sequence of models, as well as their respective AIC, ΔAIC, and AIC weights, are shown in Table 2. For full details on each model and their respective predictor variables, all statistical material is available upon request.

When comparing the different model iterations using AIC, taking mean spring water temperature, spring sunlight hours, the interaction between these two, and, finally, the marine production into account, marine 11 was the model that provided the lowest AIC and highest AIC weight of all iterations. Applying measures derived from the AIC, an ER of 4.1 was found in favor of marine 11 over marine 8, which resulted in the closest AIC value (second-lowest ΔAIC).

When examining the variable coefficients in marine 11, a statistically significant effect of summer sunlight ($P = 0.022$), as well as low $P$ values for spring water temperature ($P = 0.063$), spring sunlight ($P = 0.063$), and the interaction between spring water temperatures and sunlight ($P = 0.054$), were observed.

Models that include higher numbers of included predictor variables likely resulted in higher AIC scores due to a less parsimonious approach and potential overfitting (30).

As for the land-based AM usage models, the complexity of the collected data set was examined by PCA, based on all available predictor variables (Fig. S4). A total of seven PCs were computed, where the two most impactful PCs (1 and 2) were shown to explain 57.5% of the variation within the data set, and PC1 to PC5 were required to explain >90% of the variation. When observing the variable loadings, summer sunlight

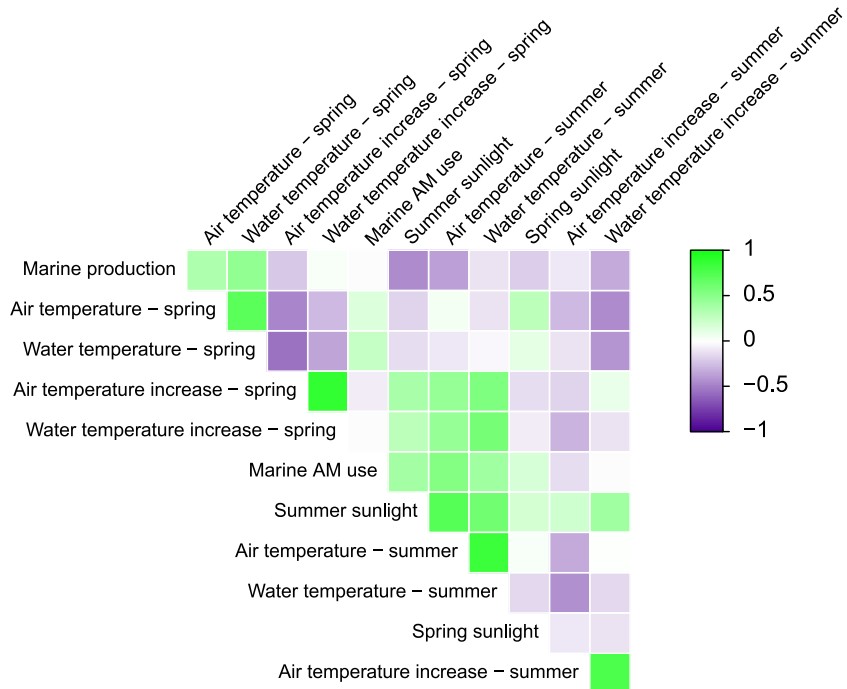

**FIG 4** Correlation matrix of data parameters for marine production. Each comparison is represented by a colored square. Parameter correlation is by color, progressing from negative correlation (purple) through no correlation (white) to positive correlation (green).

was observed to contribute directly to the variation along PC1, while spring sunlight contributed directly to the variation along PC2, supporting the marine 11 model.

## DISCUSSION

In a recent study, Schar et al. estimated that the global AM usage in aquaculture was 10,259 metric tons in 2017 and that this is expected to increase by 33% in 2030 (31). Despite the generalizing nature of their survey and the notably wide confidence

**TABLE 2** Summary of marine gamma GLM iterations[a]

| Model | Parameters | AIC | ΔAIC | AIC w |
|---|---|---|---|---|
| Marine, full | Mean water temp spring * spring sunlight * temp increase water spring + mean water temp summer * summer sunlight * temp increase water summer + marine production | 303.7 | 8.4 | 0.006 |
| Marine 1 | Mean water temp spring * spring sunlight * temp increase water spring + mean water temp summer * summer sunlight * temp increase water summer | 301.7 | 6.4 | 0.017 |
| Marine 2 | Mean water temp spring * temp increase water spring + mean water temp summer * summer sunlight * temp increase water summer | 302.1 | 6.4 | 0.014 |
| Marine 3 | Mean water temp spring + mean water temp summer * summer sunlight * temp increase water summer | 299.0 | 3.7 | 0.064 |
| Marine 4 | Mean water temp spring + mean water temp summer * temp increase water summer | 298.2 | 2.9 | 0.099 |
| Marine 5 | Mean water temp spring + temp increase water summer | 298.8 | 3.5 | 0.072 |
| Marine 6 | Mean water temp spring + temp increase water summer + marine production | 300.7 | 5.4 | 0.028 |
| Marine 7 | Mean water temp spring + marine production | 298.9 | 3.6 | 0.067 |
| Marine 8 | Mean water temp spring + mean water temp summer + marine production | 298.1 | 2.8 | 0.10 |
| Marine 9 | Mean water temp spring + mean water temp summer + summer sunlight + marine production | 298.8 | 3.5 | 0.072 |
| Marine 10 | Mean water temp spring * temp increase water spring + mean water temp summer | 299.8 | 4.5 | 0.044 |
| **Marine 11** | **Mean water temp spring * spring sunlight + summer sunlight + marine production** | **295.3** | **0.0** | **0.41** |

[a]Each iteration is listed with included model parameters. Asterisk separators denote that both connected parameters, as well as any interaction effects between them, are included in the model. The AIC for each model is given along with the ΔAIC (numerical difference between a given model AIC and the lowest AIC obtained). The model resulting in the lowest AIC is highlighted in bold.

intervals attached to the estimates proposed by Schar et al., an increased reliance on aquaculture production inevitably appears linked to a drastic increase in global AM usage. As increased AM use is generally linked to increased prevalence of resistance, this development should be counteracted from a One Health perspective. To do so, improvements on multiple fronts are required. One such is an improved understanding of how production environments affect the dynamics of pathogens and subsequent AM treatments now, as well as in the future.

Throughout the aquaculture production cycle, the environment plays an integral role in the rearing and production of aquatic animals. Considering the environment, temperature emerges as an important abiotic factor, and changes in temperature have been linked to a number of changes in immune function in a variety of fish species, as reviewed by Bowden (32). In terms of temperature and environmental influence, however, notable differences are evident between different types of production systems. Closed recirculating aquaculture system (RAS) production units offer increased control of the production environment, including temperature. Traditional land-based systems supplied with water from a local watercourse or groundwater may experience various degrees of issues in maintaining stable temperatures (33, 34). In addition, outdoor production units will still be exposed to changes in air temperature and sunshine. Marine aquaculture, on the other hand, is directly reliant on and influenced by the environment immediately surrounding the fish enclosures.

Antimicrobial use varies considerably worldwide (31). Looking to major European aquaculture-producing countries for context, Norwegian aquaculture has had an annual total AM use of 641 kg from 2010 to 2019, despite a considerably higher annual production, while AM use from the United Kingdom was a total of 160 kg in trout production and 5,600 kg in salmon production in 2020 (35, 36). In 2019, the relative AM use for Norway was 0.15 mg per population correction unit (kilograms of live weight slaughtered biomass) across all aquaculture production, while in the United Kingdom, it was 13 mg/kg and 10 mg/kg for salmon and trout production, respectively. Beyond Europe, Schar et al. estimate a worldwide mean use of 103 mg active compound per kg trout produced. Upon investigation of AM use in the present study, the variation in AM usage, whether specific or absolute, was notably higher than the variation in production. Thus, annual AM use was not strictly a function of production volume, and a closer look at additional factors beyond fluctuations in production totals was warranted.

The modeling approach of the present study was to determine the overall extent to which environmental factors, in the form of meteorological parameters, could help, at least partly, explain variability in annual AM use, based on data from Danish aquaculture. A modeling study such as this represents a broad approach, and it is important to note that the meteorological parameters are to be considered indirect, nonexclusive reasons for antimicrobial usage. The extent and severity of disease outbreaks that require AM treatments in aquaculture are likely caused by a combination of several factors. Apart from environmental factors, factors like vaccination status and efficacy, feeding programs, as well as stocking densities and handling of the fish, which likely impose stress on the fish and thereby potentially affect disease resistance, are also expected to influence AM use (Fig. 5). While assessment of the combined effects of the factors summarized in Fig. 5 is beyond the scope of the present study, previous reports have demonstrated clear effects of efficient vaccine programs in aquaculture as an example of the potential impact of just one of the factors in Fig. 5 (37, 38). Vaccine use would therefore have been an ideal parameter to include in the current study, as it would be expected to have a substantial effect, and future research projects should include vaccine data collection from all or subsets of Danish aquaculture sites, considering vaccine coverage and route of administration (injection, immersion, etc. [39]) to enable additional insights into the fluctuations observed in annual AM use.

Having mined data from publicly available repositories, the nature and potential interconnectivity of the data variables were evaluated, and finally, relevant data variables with limited degrees of intercorrelation were included in the proposed models.

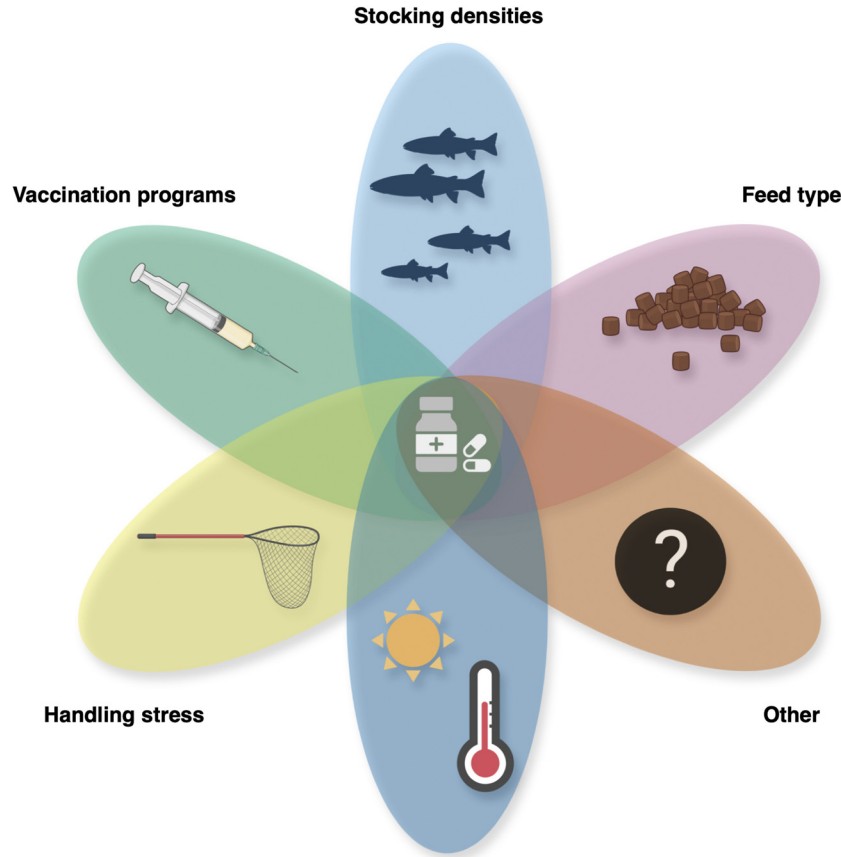

**FIG 5** Graphical representation of factors expected to affect AM use. The fluctuation in AM usage that lies at the center of this study is likely affected by multiple factors. These include vaccine use and efficacy, stocking densities, feeding programs, as well as various external stressors, such as handling stress and environmental stressors, that are beyond the scope of this study. The size of each ellipsoid is arbitrary and thus does not reflect relative suspected impacts.

Once the two individual families of nested models were constructed, the AIC score provided a means of ranking the individual models. The scores in one family of models are, however, not intended to translate to other series of models (40). Consequently, land-based and marine models should not be compared based on AIC score.

Using AIC, along with AIC weight, predictor variable significance, and evidence ratio, one model was identified as providing the best fit to the data and thus highlighted some of the driving forces behind the observed variation in annual AM usage. While only one model comes in first place, valuable insights can be gained from observing the results from contending models with low ΔAIC.

By applying an AIC-guided selection approach to the data presented here, summer sunlight hours and annual production (land 3), as well as summer air temperature (land 2, land 4, and land 5) were identified as the most likely factors to influence the variation in annual AM use in land-based production. It should be noted that while summer sunlight hours and summer air temperature both were identified as influential factors independently, neither of them retained statistical significance when applied in a model together. This could be due to the observed potential collinearity between the two. The close relation and effect on annual AM use of these two factors were supported further by the variable loadings in the principal-component analysis. Finally, the link between these two parameters also appears plausible from a logical standpoint, and while the summer sunlight hours parameter provided a slightly better fit to the data than the summer air temperature (land 3 versus land 5), the relative importance of these two parameters was hard to

distinguish. The influence of temperature and summer sunlight is in line with previous findings from Danish freshwater aquaculture which demonstrated that while rainbow trout infected with *Flavobacterium psychrophilum* were typically found during colder months, consistent with its reported severity at ≤15℃, *Y. ruckeri* and *A. salmonicida* infections were primarily observed during the warmer summer months (41, 42).

A more complicated picture emerged from the marine models. Based on the best fit to the data set, summer sunlight, spring water temperature, spring sunlight, the interaction between spring water temperatures and sunlight, as well as annual production, were identified as influential data parameters. Furthermore, mean summer water temperature was found to be of notable, albeit not statistically significant, influence ($P = 0.09$) in the model with the second-lowest $\Delta$AIC, marine 8.

Interestingly, while summer sunlight hours and air temperatures were found to be notably correlated (PCC = 0.74) when modeling land-based production, this was not observed to be the case for spring sunlight and spring water temperatures (PCC = 0.11). This limits the risk of collinearity, and the importance of the observed interaction in marine 11 appears less likely to be an artifact.

Apart from affecting fish immune reactions, temperature has also been suggested to affect the virulence of bacterial fish pathogens. This is reflected in a previous descriptive study from Danish marine aquaculture, where *A. salmonicida*, *Photobacterium damsela*, *V. anguillarum*, and *Vibrio parahaemolyticus* were primarily isolated from rainbow trout during summer months when recorded water temperatures were higher, which is in line with the AM treatment records included in the present study (43). On a meta level, a recent modeling study investigating the extent of multiantibiotic-resistant bacteria and the influence of climate change on the spread of such organisms revealed a general positive correlation between temperature and mortality following bacterial infection in both temperate and tropical/subtropical environments (44). The study also suggested that temperature was significantly, positively correlated with the presence of antimicrobial resistance (AMR) genes. The study, published by Reverter et al., included several relevant genera, such as *Yersinia*, *Aeromonas*, *Flavobacterium*, and *Vibrio*, and while mortality following infection with *Y. ruckeri* was not observed to increase significantly with temperature, the general trend was clear, stating an expected ≈4 to 6% increase in mortality following a 1℃ increase in temperature in temperate environments. Thus, even given a successful effort to limit global temperature increase to less than 2℃ above preindustrial levels as declared in the Paris Agreement (45), a rise in AMR gene frequency and elevated mortality rates in aquaculture may be expected.

In conclusion, the results from the modeling approaches to both land-based and marine fish productions show that models based on meteorological parameters, along with annual production totals, were successfully fitted to the annual AM usage. Further, the proposed models, at least in part, explained the observed variations in AM usage. Through the refinement of these models, summer sunlight hours were observed to have a statistically significant impact on AM use for both land-based and marine aquaculture. This was in accordance with one of the rules of thumb that formed the base hypotheses of this study. Further, the optimal model for marine AM use relied on effects of spring sunlight and water temperatures, that, while not statistically significant, lends support to the other rule of thumb that increased water temperatures contribute to increasing AM use.

The conclusions presented in this study can help to identify periods during production cycles where preventive measures, including pulse-feeding of probiotic supplements and lowered stocking densities, potentially could help reduce infection pressure and resulting AM usage (46). Furthermore, they provide an insight into the potential effects of temperature increases, not only in Danish, but also in production systems around the world affected by similar environmental and abiotic factors.

## MATERIALS AND METHODS

**Aquaculture production data.** Data on annual land-based and marine aquaculture production (metric tons/year, $n = 19$) were obtained with assistance from the Danish aquaculture organization Dansk Akvakultur (Danish Aquaculture [DA]). The production data come from two separate original

sources. Production totals for land-based and marine aquaculture from 2001 until 2008 were from the Danish Fisheries Agency. The statistics cover land-based production (freshwater farms and recirculated farming systems), as well as marine production (mariculture systems). From 2009 to 2019, production totals originated from Statistics Denmark, a national statistics database. The data sets cover land-based production (freshwater farms, model farm systems, recirculated systems, land-based seawater farms, eel farms, as well as farm types designated "other") and marine production (strictly mariculture systems). Shellfish production is not considered in this study.

**Antimicrobial usage data.** Data on annual use of antimicrobials (kilograms active substance/year) from 2001 to 2019 were obtained from DA on the basis of annual reporting to the Danish veterinary authorities. These data set was matched against the annual, publicly available DANMAP reports from 2001 to 2019 (1, 2, 12–28). An average annual difference of 13.8 kg active substance (standard deviation [SD], 91.9 kg) was identified between the two data sets, resulting in an average annual difference of 0.4% (SD, 3.2). The data from DA allow for a differentiation between freshwater and marine aquaculture, while the DANMAP data only list the total annual AM prescriptions. Therefore, the data from DA were used for the modeling approaches presented in this study. DANMAP report data allowed insight into the relative use of each class of AM. Annual vaccine sales figures were also provided by DA.

**Meteorological data.** To evaluate the effects of meteorological parameters, data were obtained from archival databases at the Danish Meteorological Institute (DMI). The databases contain seasonal weather data relating to all four seasons, with spring defined as March, April, and May and summer defined as June, July, and August (47). The data used in this study include the reported nationwide mean temperature and total recorded hours of sunshine per month for the spring and summer months, respectively. The data are publicly available from the DMI (47–49). In addition, monthly water temperatures from the spring and summer months were retrieved using DMI Open Data API access as part of the oceanographic observation (OceanObs) data set. The water temperature data set features round-the-clock measurements and cover Danish waters broadly (approximately 140 measuring stations). The OceanObs data are raw, unprocessed data and have not been subjected to internal quality control measures by the DMI. However, while occasional measurements could thus potentially be erroneous, the high total number of data points is expected to effectively outweigh such errors.

Increases in both air and water temperature were subsequently calculated by linear regression on monthly temperature means for both spring and summer months to obtain a monthly increase estimate across each season.

**Statistical analyses.** Raw data were curated using Microsoft Excel for Mac, version 15.69. All statistical analyses were performed in R version 4.0.3, Bunny-Wunnies Freak Out (50), through R Studio version 1.4.1103. The following packages were used: readxl (51), ggplot2 (52), dplyr (53), patchwork (54), ggpubr (55), plotly (56), MASS (57), goft (58), ggbiplot (59), qpcR (60), and corrr (61).

A complete rundown of the performed analyses is available upon request. Figures were created in R, except for Fig. 5, which was created using BioRender.com and Keynote for Mac, version 11.2.

**Statistical modeling.** To assess whether variation in meteorological parameters could explain part of the variation observed in the AM usage, statistical models were constructed with AM usage as the outcome, or response variable, and meteorological parameters, as well as production data, as predictor variables influencing the response variable. The absolute AM use was chosen over the relative AM use to avoid a ratio as response variable.

Given that not all study parameters are applicable to both production forms, land-based and marine productions were modeled individually. Before the models were set up, an appropriate data distribution was investigated. As positive, continuous outcomes were base requirements for both response variables, and as a slight tendency toward a positive skew of the plotted frequency distributions was observed, both were suspected to follow an underlying gamma distribution (62). For both response variable data sets, this was addressed by visual observation of plotted data versus a theoretical distribution, as well as a variance ratio test, and found to be appropriate (63). Thus, for both production forms, a gamma generalized linear model (GLM) approach was chosen. As a tendency toward increased scatter with increasing predicted values was observed, a logarithmic link function was included in the GLMs to avoid issues with uneven errors in the statistical models.

From this point on, land-based and marine productions were treated and modeled independently.

For both production forms, Pearson correlations between predictor value data were performed to identify any correlations, or collinearities, between predictor variables prior to establishing the models (Data available upon request). An initial "full" model then was set up utilizing all available predictor variables, as well as relevant interaction links. Based on the result of the "full" model, manually curated iterations of nested models were adapted and evaluated. All model iterations for each production form were based on the same data set and were thus considered nested. As such, each iterative adaptation of a model was evaluated based on Akaike information criteria (AIC), ΔAIC, AIC weight, evidence ratio (ER), and statistical significance (40, 64). The AIC allows the comparison of models, taking model fit, as well as the number of model parameters, into account. While improving the maximum likelihood of the model fit to the data, each inclusion of a new parameter was penalized in the score. This was done to avoid overfitting by ensuring that any extra model parameter needed to improve the model fit beyond the penalty added for its inclusion. When comparing nested models based on the same data set, a lower AIC indicated a better fit for the model in question, based on the specific model parameters chosen.

$$\text{AIC}_i = -2\log L_i + 2V_i \tag{1}$$

for model $i$, where $L_i$ is the maximum likelihood obtained by adjusting $V_i$ model parameters. Note that a penalty was added for each included model parameter.

$$\Delta \text{AIC}_i \quad = \text{AIC}_i - \text{minAIC} \tag{2}$$

for model $i$, where $\text{AIC}_i$ is the calculated AIC for model $I$ (see equation 1), and min AIC is the numerically lowest calculated AIC for the family of models in question.

$$\text{AIC} \quad \text{weight}_i = \frac{\exp[-0.5 \times \Delta_i \ (\text{AIC})]}{\sum_{k=1}^{K} \exp[-0.5 \times \Delta_i \ (\text{AIC})]} \tag{3}$$

where $\exp[-0.5 \times \Delta_i \ (\text{AIC})]$ expresses the relative likelihood of model $I$, and $\sum_{k=1}^{K} \exp[-0.5 \times \Delta_i \ (\text{AIC})]$ expresses the sum of relative likelihoods of all models $k$.

$$\text{ER} = \frac{\text{AIC} \quad \text{weight}_{\text{high}}}{\text{AIC} \quad \text{weight}_{\text{low}}} \tag{4}$$

where AIC weight$_{\text{high}}$ refers to the numerically higher of two AIC weights being compared, and AIC weight$_{\text{low}}$ refers to the lower.

When applicable, an $\alpha$-level of 0.05 was used for hypothesis testing. A $P$ value of $<0.05$ caused the alternative hypothesis to be favored over the null hypothesis that the parameter in question did not significantly influence the outcome of the response variable.

Model refinement was performed manually, favoring the improvement of AIC and statistical significance in each iteration.

**Data availability.** All data included in this study are publicly available through the cited sources. All statistical material is available upon request.

## SUPPLEMENTAL MATERIAL

Supplemental material is available online only.

**SUPPLEMENTAL FILE 1**, PDF file, 0.2 MB.

## ACKNOWLEDGMENTS

We thank senior advisor Anette Ella Boklund, University of Copenhagen, for providing valuable sparring and input regarding data modeling and Niels-Henrik Henriksen and Dansk Akvakultur for providing data on AM use and for relaying observations from aquaculture professionals.

A.M.B. and K.R.V. conceptualized the study and framed the research questions, K.R.V. collected and analyzed the data, A.M.B. and K.R.V. discussed and summarized the outcome of the analyses, K.R.V. wrote the manuscript, and A.M.B. and K.R.V. discussed and agreed upon the final manuscript and its submission.

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
