## [Reviewer comments · Microbiology Spectrum]

Microbiology Spectrum

Addressing a Rule of Thumb - Modelling the Effects of Meteorological Conditions on Prescription of Antimicrobials in Aquaculture

Kasper Rømer Villumsen and Anders Bojesen

Corresponding Author(s): Anders Bojesen, University of Copenhagen

Review Timeline:

Submission Date:	May 11, 2022
Editorial Decision:	August 17, 2022
Revision Received:	August 31, 2022
Accepted:	September 1, 2022

Editor: Artem Rogovskyy

Reviewer(s): The reviewers have opted to remain anonymous.

Transaction Report:

DOI: <https://doi.org/10.1128/spectrum.01752-22>

August 17, 2022

Prof. Anders Miki Bojesen
University of Copenhagen
Veterinary and Animal Science
Stigbøjlen 4,
Frederiksberg C DK-1870
Denmark

Re: Spectrum01752-22 (Addressing a Rule of Thumb - Modelling the Effects of Meteorological Conditions on Prescription of Antimicrobials in Aquaculture)

Dear Prof. Anders Miki Bojesen:

Link Not Available

Sincerely,

Artem Rogovskyy

Journals Department
Reviewer comments:

Reviewer #1 (Comments for the Author):

This manuscript described any relationship between environmental temperature and the use of antimicrobials (AM). To be honest I am not qualified to evaluate the mathematical modeling and the statistical methods used in this study. But, I have reviewed the other parts. Please consider the following points in the revision:

Line 99: n=19. Not clear - please detail.

Line 218: Please check the AM use. The authors sate gram per kilo weight. Do you mean body weight. It appears that the AM

use is thus quite excessive compared with e.g., Norwegian aquaculture.

General: It would have been interesting to see the AM use in:

1. Marine-based systems
2. Land-based recirculation system
3. Land-based flow through system with ground water supply
4. Land-based system with supply of river/lake water

This would pinpoint where any effort should be put to reduce the use of AM. Please consider to add data (in supplement).

In general: Could the authors give a couple of more sentences to compare the use of AM between relevant aquaculture countries. Is the Danish AM use comparable with e.g., Scottish, Faroe Island, Norwegian countries.

Figure 1: Much more detailed figure caption has to be made. Please mark figures with A, B and C. Query: gram per kilo - is this correct?

I feel that Figure 5 for me ok (-), but too speculative for many readers, to be in the main text. Please consider to have this figure as supplement, as there are uncertainties how much each factors contributes to AM use.

Reviewer #2 (Comments for the Author):

I cannot see the mechanistic link between meteorological factors and antimicrobials. The authors require to understand the real environmental conditions in which aquaculture is carried out, the diseases and antibiotic use. The simplistic presentation of the authors of the problem analyzing few meteorological factors seems to have a very low power beyond the statistics to allow to reach relevant and realistic conclusions. In addition, the MS focus on local data (Danish) and is not easy to get a broader analysis on the subject, do not use the existing knowledge in other countries and does not use the existing international literature on antimicrobial use and aquaculture development.

Reviewer #3 (Comments for the Author):

Overall this is an exceptionally well written paper, addressing a a clear and interesting question, conducted to a very high quality. There are no major issues that need addressing, but I have a few thoughts:

1. Abstract L15-16. Please can you be more specific and precise about the two main results (the factors in the best fit models for marine and land systems) in your description of the results.
2. Abstract: Some readers would also like to see quantification in the abstract (e.g. p values or AIC values); I make students do it but am personally less bothered in 'actual' manuscript abstracts; I leave the choice to the discretion of the editor.
3. Figures 2 and 4. While I appreciate the creativity in showing these data in this way, these figures are somewhat unusual. Have the authors considered a more 'standard' approach, that is, to display the correlations as a correlation matrix, coloured accordingly? (I would normally cluster the matrix too so that correlated variables appear close to each other, but not show any underlying trees, as the purpose of the clustering would be cosmetic). I would find such an approach easier to parse, but that is a matter of taste rather than correctness. I would suggest that the authors produce alternative figures as I have suggested, and then see which they prefer - if that is what they have then fine.
4. L251-252. Please be more precise in describing the Land3 model in the text of the Results so it can be fully understood without reference to the table.

Staff Comments:

Preparing Revision Guidelines

Please return the manuscript within 60 days; if you cannot complete the modification within this time period, please contact me. If you do not wish to modify the manuscript and prefer to submit it to another journal, please notify me of your decision immediately so that the manuscript may be formally withdrawn from consideration by Microbiology Spectrum.

Response to reviewers

Please find our point-by-point responses below. The following contains the review comments by all three reviewers, with our responses included in *italics*. For a detailed overview of all changes made, please see the submitted mark-up .docx-file.

Reviewer #1 (Comments for the Author):

This manuscript described any relationship between environmental temperature and the use of antimicrobials (AM). To be honest I am not qualified to evaluate the mathematical modeling and the statistical methods used in this study. But, I have reviewed the other parts. Please consider the following points in the revision:

Line 99: n=19. Not clear - please detail.

- We agree, and the line in question has now been changed to (Line 116 in the revised manuscript with markup): "Data on annual use of antimicrobials (kg active substance/year) from 2001-2019 were obtained from DA, on basis of annual reporting to the Danish veterinary authorities.". Hopefully, this has improved the clarity of the data foundation.

Line 218: Please check the AM use. The authors state gram per kilo weight. Do you mean body weight. It appears that the AM use is thus quite excessive compared with e.g., Norwegian aquaculture.

- This was an error on our side. The absolute annual AM use is in kg, and the relative AM use is in gram/ton. Not gram/kg as previously written. This has now been corrected throughout the manuscript.

General: It would have been interesting to see the AM use in:

1. Marine-based systems
2. Land-based recirculation system
3. Land-based flow through system with ground water supply
4. Land-based system with supply of river/lake water

This would pinpoint where any effort should be put to reduce the use of AM. Please consider to add data (in supplement).

- Absolutely, however we are limited by the way that AM use is reported. While the collection of AM data is very thorough, this is the partitioning of data that we had to work with.

In general: Could the authors give a couple of more sentences to compare the use of AM between relevant aquaculture countries. Is the Danish AM use comparable with e.g., Scottish, Faroe Island, Norwegian countries.

- This is a good point. For context, AM use from Norwegian and UK aquaculture has now been introduced in the discussion, as follows (Lines 373-380 in the revised marked up manuscript):

“Antimicrobial use varies considerably worldwide (1). Looking to major European aquaculture producing countries for context, Norwegian aquaculture has had an annual total AM use of 641 kg from 2010-2019, despite a considerably higher annual production, while AM use from the United Kingdom were a total of 160 kg in trout production and 5600 kg in salmon production in 2020 (2, 3). In 2019 the relative AM use for Norway was 0.15 mg per population correction unit (kg live-weight slaughtered biomass) across all aquaculture production, while in the United Kingdom it was 13 mg/kg and 10 mg/kg, for salmon and trout production respectively. Beyond Europe, Schar et al estimate a worldwide mean use of 103 mg active compound per kg trout produced.”

Figure 1: Much more detailed figure caption has to be made. Please mark figures with A, B and C. Query: gram per kilo - is this correct?

- Uppercase letters have now been added to the figures. The legend was also updated to match the rest of the document. The legend has now been changed to “Figure 1: Overview of production, AM use and AM use relative to production for Danish aquaculture from 2001-2019. A) Annual production (tons), B) absolute AM use (kg) and C) relative AM use (g/ton) in land-based (green) and marine (blue) production.”

I feel that Figure 5 for me ok (-), but too speculative for many readers, to be in the main text. Please consider to have this figure as supplement, as there are uncertainties how much each factors contributes to AM use.

- We appreciate this suggestion. However, as evident from comments made during this review, we feel that it is very important to note the influences of stressors and production environment parameters that are not considered in this study. We therefore consider this figure quite important to the discussion of the results that we present and prefer to keep the figure in the main text. This suggestion does raise an important point about the relative impacts of each factor. To avoid any confusion regarding the arbitrary sizes of the figure components, the legend has now been changed to the following:

“Figure 5: A graphical representation of factors expected to affect AM use. The fluctuation in AM usage that lies at the center of this study, is likely affected by multiple factors. These include vaccine use and -efficacy, stocking densities, feeding programs, as well as various external stressors, such as handling stress and environmental stressors, that are beyond the scope of this study. The size of each ellipsoid is arbitrary, and thus does not reflect relative suspected impacts.”

Reviewer #2 (Comments for the Author):

I cannot see the mechanistic link between meteorological factors and antimicrobials. The authors require to understand the real environmental conditions in which aquaculture is carried out, the diseases and antibiotic use. The simplistic presentation of the authors of the problem analyzing few meteorological factors seems to have a very low power beyond the statistics to allow to reach relevant and realistic conclusions. In addition, the MS focus on local data (Danish) and is not easy to get a broader analysis on the subject, do not use the existing knowledge in other countries and does not use the existing international literature on antimicrobial use and aquaculture development.

- We respectfully disagree with reviewer 2. We acknowledge that the aquaculture production, like any other production form, is highly complex and that no single factor can explain variations in eg. AM use. For that exact reason, we have been careful not to claim that the mechanisms described in

this manuscript are the sole reasons for the observed fluctuations. We simply wanted to investigate if there were patterns in historic meteorological data that correlated with AM usage, and if any such patterns could plausibly have an effect on treatment-requiring outbreaks of bacterial pathogens. We believe that we have clearly stated that any weather-related mechanisms will be part of a larger interplay between several other stressors, prophylactic strategies and environmental conditions, as shown in figure 5. As requested by reviewers 1 and 2, contextual data regarding AM use in other countries have been included in the manuscript.

Reviewer #3 (Comments for the Author):

Overall this is an exceptionally well written paper, addressing a a clear and interesting question, conducted to a very high quality. There are no major issues that need addressing, but I have a few thoughts:

- *We appreciate the effort and the comments made.*

1. Abstract L15-16. Please can you be more specific and precise about the two main results (the factors in the best fit models for marine and land systems) in your description of the results.

- *Yes we can. Without introducing any P- or AIC-values (see point 2 below), we have changed the sentence to the following (lines 15-19 in the revised manuscript):*

“Separate series of models were made and refined for land-based and marine production, respectively. For both production forms the models identify summer sunlight hours as having a significant influence on antimicrobial use. In addition to summer sunlight, spring sunlight and water temperature were integral, although not statistically significant parameters when modelling antimicrobial use in the marine production.” Hopefully this has improved the specificity sufficiently.

2. Abstract: Some readers would also like to see quantification in the abstract (e.g. p values or AIC values); I make students do it but am personally less bothered in 'actual' manuscript abstracts; I leave the choice to the discretion of the editor.

- *We appreciate the suggestion. Given the nature and complexity of the model results, however, we feel that adding P- and AIC-values to the abstract would make it an unnecessarily heavy and complex read. We would prefer to keep it as is if this is acceptable.*

3. Figures 2 and 4. While I appreciate the creativity in showing these data in this way, these figures are somewhat unusual. Have the authors considered a more 'standard' approach, that is, to display the correlations as a correlation matrix, coloured accordingly? (I would normally cluster the matrix too so that correlated variables appear close to each other, but not show any underlying trees, as the purpose of the clustering would be cosmetic). I would find such an approach easier to parse, but that is a matter of taste rather than correctness. I would suggest that the authors produce alternative figures as I have suggested, and then see which they prefer - if that is what they have then fine.

- *We followed the suggestion and produced alternative correlation matrices. In our opinion, they did the job better, and we have now changed the figure layouts for figures 2 and 4. Thank you for very valuable input. The figure legends have been changed to match the new layout:*

“Figure 2: Correlation matrix of data parameters for land-based production. Each comparison is represented by a colored square. Parameter correlation is by color, progressing from negative correlation (purple) through no correlation (white) to positive correlation (green).”

“Figure 4: Correlation matrix of data parameters for marine production. Each comparison is represented by a colored square. Parameter correlation is by color, progressing from negative correlation (purple) through no correlation (white) to positive correlation (green).”

4. L251-252. Please be more precise in describing the Land3 model in the text of the Results so it can be fully understood without reference to the table.

- We appreciate this suggestion. We have tried to improve the description of the Land3 model within the existing framework. The description in question now reads (Lines 272-277 in the revised marked up manuscript):

“Using AIC as a guide, the model Land3 posted the lowest AIC and the highest AIC weight. The Land3 model incorporates summer sunlight and annual land-based production totals as predictor variables, with no interaction links. The model output for Land3 (see supplementary material for full output), indicated that the only the summer sunlight variable had a statistically significant impact on the response variable, the annual AM use ($P=0.0011$, for definition of null and alternative hypotheses, see Data Foundation and Methods).”

Additional change:

Due to the very large size and page number of the supplementary statistical file, we have decided to remove it from the submission, but to make it freely available upon request.

1. Schar D, Klein EY, Laxminarayan R, Gilbert M, Van Boeckel TP. 2020. Global trends in antimicrobial use in aquaculture. *Scientific Reports* 10.
2. NORM, NORM-VET. 2020. Usage of Antimicrobial Agents and Occurrence of Antimicrobial Resistance in Norway.
3. UK-VARSS. 2021. Veterinary Antibiotic Resistance and Sales Surveillance Report.

September 1, 2022

Prof. Anders Miki Bojesen
University of Copenhagen
Veterinary and Animal Science
Stigbøjlen 4,
Frederiksberg C DK-1870
Denmark

Re: Spectrum01752-22R1 (Addressing a Rule of Thumb - Modelling the Effects of Meteorological Conditions on Prescription of Antimicrobials in Aquaculture)

Dear Prof. Anders Miki Bojesen:

Your manuscript has been accepted, and I am forwarding it to the ASM Journals Department for publication. You will be notified when your proofs are ready to be viewed.

Sincerely,

Artem Rogovskyy
Editor, Microbiology Spectrum
